# Accurate Prediction of Cancer Prognosis by Exploiting Patient-Specific Cancer Driver Genes

**DOI:** 10.3390/ijms24076445

**Published:** 2023-03-29

**Authors:** Suyeon Lee, Heewon Jung, Jiwoo Park, Jaegyoon Ahn

**Affiliations:** 1Department of Computer Science and Engineering, Incheon National University, Incheon 22012, Republic of Korea; 2Samsung Electronics Company Ltd., Suwon 16677, Republic of Korea

**Keywords:** cancer prognosis, cancer driver gene, machine learning

## Abstract

Accurate prediction of the prognoses of cancer patients and identification of prognostic biomarkers are both important for the improved treatment of cancer patients, in addition to enhanced anticancer drugs. Many previous bioinformatic studies have been carried out to achieve this goal; however, there remains room for improvement in terms of accuracy. In this study, we demonstrated that patient-specific cancer driver genes could be used to predict cancer prognoses more accurately. To identify patient-specific cancer driver genes, we first generated patient-specific gene networks before using modified PageRank to generate feature vectors that represented the impacts genes had on the patient-specific gene network. Subsequently, the feature vectors of the good and poor prognosis groups were used to train the deep feedforward network. For the 11 cancer types in the TCGA data, the proposed method showed a significantly better prediction performance than the existing state-of-the-art methods for three cancer types (BRCA, CESC and PAAD), better performance for five cancer types (COAD, ESCA, HNSC, KIRC and STAD), and a similar or slightly worse performance for the remaining three cancer types (BLCA, LIHC and LUAD). Furthermore, the case study for the identified breast cancer and cervical squamous cell carcinoma prognostic genes and their subnetworks included several pathways associated with the progression of breast cancer and cervical squamous cell carcinoma. These results suggested that heterogeneous cancer driver information may be associated with cancer prognosis.

## 1. Introduction

The accurate prediction of the prognoses of cancer patients is important since it can allow the provision of improved treatment and help design anticancer drugs by enhancing our understanding of cancer progression. Numerous bioinformatics studies have previously been conducted for the accurate prediction of cancer prognoses and the identification of prognostic biomarkers. These studies were primarily focused on developing statistical [1] or machine learning methods [2,3] and then applying them to various types of omics data. Among these two approaches, machine learning methods have been gaining increasing attention recently, and have shown good performances as a result of recent advances in machine learning, such as deep learning, and the accumulation of omics data.

Bioinformatics studies for the prediction of cancer patients can be divided roughly into those that are focused on each gene independently and those that consider the relationship between genes. The latter category uses genetic network data, such as the protein–protein interaction network (PPI) and the gene regulation network [4,5], alongside omics data. The exploitation of genetic network data is advantageous since it provides a better understanding of cancer development and progression, considering that prognostic genes can be identified at the genetic network level. Additionally, this can reduce problems with dimensionality that are caused by having few samples compared to numerous genes. Therefore, machine-learning methods can benefit from the use of genetic network data.

The majority of state-of-the-art methods for predicting cancer prognosis involve the application of machine learning methods, including deep learning, with both multi-omics and genetic network data. For example, GVES [6] applies the CBOW model [7] to a sequence of genes obtained by random walk on a functional interaction network to predict prognostic genes. Furthermore, GEDFN [8] incorporates genetic networks into deep neural networks. The incorporation of a genetic network prevents overfitting with sparse connections between the layers. Subsequently, the importance scores of the features were calculated using the extended CW method [9]. The model for predicting prognoses using a GAN [10] reconstructed the FI network using multiple omics data and applied the network for the connection of the generator layers. KatzDriver [11] first combined gene expression data and gene regulatory networks to calculate the weights of nodes and edges, before calculating the relative influence of each gene based on the Katz [12] algorithm and identifying the top gene with the highest relative influence ranking as a biomarker. In addition, DeepProg [13] uses a boosting strategy in which autoencoders transform multi-omics data, and each hidden layer is used to test its association with survival with multiple models. Afterwards, the correlation was calculated for the gene pair, while a gene regulation network was constructed to cluster and predict prognoses using SVM.

While most of the previously described methods use gene expression or other omics data to predict cancer prognoses, this study focuses on the potential use of genes as a driver gene to predict cancer prognoses, considering that it is very likely that patients with different prognoses will have different driver genes. To calculate the likelihood of each gene being a driver gene, DawnRank [14], which is a patient-specific cancer driver gene identification method, is used here. For more accurate prediction, we apply DawnRank to the patient-specific gene network that was created with differences in gene expression between cancer and normal samples for each patient and adjust the gene score. We then input the adjusted scores of genes that showed significantly different expression between the good and bad prognosis groups into the deep feed-forward network to predict the prognosis.

Applying the proposed method to 11 cancer types from TCGA [15] data showed a significantly improved prediction performance compared to the existing state-of-the-art methods for three cancer types (BRCA, CESC and PAAD), better performance for five cancer types (COAD, ESCA, HNSC, KIRC and STAD), and similar or slightly worse performance for the remaining three cancer types (BLCA, LIHC and LUAD). Functional analysis of the identified prognostic genes for breast cancer and cervical squamous cell carcinoma was performed, with many genes being included in known pathways associated with the progression of breast cancer and cervical squamous cell carcinoma. These results showed that the driver gene score was better for prognosis prediction than the expression itself, implying that cancer drivers were heterogeneous for poor and good prognosis groups.

## 2. Results

### 2.1. Data Description

Here, 11 cancers were selected from TCGA [15] that had both tumor and normal samples, mutation information, and clinical information. These cancer types included bladder urothelial carcinoma (BLCA), invasive breast carcinoma (BRCA), cervical squamous cell carcinoma, endocervical adenocarcinoma (CESC), colon adenocarcinoma (COAD), esophageal carcinoma (ESCA), head and neck squamous cell carcinoma (HNSC), kidney renal clear cell carcinoma (KIRC), liver hepatocellular carcinoma (LIHC), lung adenocarcinoma (LUAD), pancreatic adenocarcinoma (PAAD), and stomach adenocarcinoma (STAD). Gene expression, clinical data, and somatic mutation data were then downloaded from the TCGA database. In the gene expression data, genes with an expression value of zero in more than 80% of the sample were excluded.

A gene network was also collected and integrated using a functional interaction (FI) network from Reactome [16], in addition to a gene regulation network from RegNetwork [17] and TRRUST [18]. The number of genes/edges in the FI network, RegNetwork, and TRRUST were 14,071/110,721, 23,336/372,774, and 2852/9383, respectively. Furthermore, the number of genes and edges in the integrated network was 25,167 and 490,200, respectively.

Based on the patient’s death information and survival period from the clinical data, the prognosis was predicted to be bad if the patient died before the criteria year, and good if not. Cancer types have different criteria over time, with BLCA, BRCA, KIRC, LIHC, LUAD, PAAD and STAD being the known criteria. For other cancer types, we set a criterion year that balanced the number of good and bad samples in the gene expression data. Table 1 shows the data for each cancer sample obtained from the TCGA database.

### 2.2. Model Configuration

To demonstrate that gene selection based on *p*-values was appropriate, the performance of the DNN was compared using *p*-value-based genes, all genes, KEGG genes [19], and randomly selected genes. After generating good and bad samples, the training and test sets were randomly divided using a ratio of 5:5. In the case of the *p*-value, 5-fold cross-validation was performed on the training set. Genes were then selected using [0.1, 0.05, 0.01, 0.005] as the thresholds for each fold change. Thereafter, the model was constructed by selecting the *p*-value with the highest average AUC of 5-fold tests, with the results being derived using the test set. The *p*-value selected for each cancer or configured training set was varied; therefore, the *p*-value was not fixed separately, and the optimal *p*-value was determined according to the situation. The number of genes in the method of randomly selecting genes was selected based on the number of genes selected using the *p*-value. Finally, as shown in Figure 1, the genes selected based on *p*-values showed the highest performance.

Next, to demonstrate that the proposed patient-specific gene network was suitable, its performance was compared using the proposed patient-specific gene network, a randomly weighted network, and an unweighted network, as in Dawnrank. As shown in Figure 2a, the average AUC of 10 random samples was the highest for the patient-specific network in most carcinomas. Although the AUC of randomly weighted networks was higher in PAAD, the use of patient-specific networks provided stability in terms of prediction by showing significantly smaller deviations in patient-specific networks across all cancers.

The results were also compared using the gene score adjustment scheme, the gene score without win rate, and the gene scores as inputs. Figure 2b shows that the adjustment using the win rate was critical for predicting cancer prognosis.

### 2.3. Hyperparameter Tuning

The hyperparameters adjusted in the DNN model were the learning rate and batch size. Other model configurations had three hidden layers, with ReLU being used between the hidden layers as the activation function, while Sigmoid was used for the output layer. The total number of epochs was 200, although learning was terminated if a loss value of less than 0.0001 was accumulated more than 20 times during learning. For comparison, the AUC of the DNN model was measured using the LIHC sample. The gene was selected based on the p-value with the highest average AUC in the five folds of the training set. After training the entire training set with the selected gene, the test set was evaluated. This method was subsequently repeated a total of 10 times. Figure 3 shows that the average AUC was highest when the learning rate was 0.001 and the batch size was 2. When the learning rate was 0.0001, the AUC remained similarly high, although it took more than twice as long as when at 0.001, so this was not selected.

We also compared various machine learning methods such as Random Forest, XGBoost [20], LightGBM [21], CatBoost [22], and DNN. In Figure 4, DNN shows a significantly higher AUC, so we selected DNN as our classifier.

### 2.4. Comparison on Different Machine Learning Methods

To evaluate the performance of the proposed method, the AUC, PR-AUC, balanced accuracy, F1-score, and Matthews correlation coefficient values of the CPR, GEDFN, Wu and Stein [23], WGCNA, GVES, and DeepProg were compared. In most cancers, except for BLCA, LIHC, and LUAD, the proposed method demonstrated a superior average performance compared to the majority of previous studies, as shown in Figure 5 and Appendix A. The proposed method also significantly outperformed in cases of BRCA, CESC, and PAAD. DeepProg showed an excellent average performance in most cancers, especially BLCA, LIHC, and LUAD; however, the accuracies were not significant compared to the proposed method and showed a large deviation compared to the proposed method. The low deviation of the proposed method indicated that stable prediction was possible through the generation of patient-specific impact vectors.

### 2.5. Functional Analysis of Prognostic Genes

The proposed method exhibited significantly higher AUC values for BRCA, CESC, and PAAD. A functional analysis was then performed for the prognostic genes BRCA and CESC, which commonly affect women.

To select prognostic genes for BRCA, genes were initially selected with a final score (win rate) in the top 30 for ten experiments. As a result, 25 genes were identified thrice out of ten experiments and were subsequently selected as prognostic genes (Appendix A). Eleven of the twenty-five prognostic genes were selected more than four times, while six genes were known BRCA driver genes (APC, BRCA1, MAX, RB1, RUNX1, and SMAD2) in the CGC [24] and Intogen [25] databases. A subgraph of the 25 prognostic genes is shown in Figure 6. Using patient-specific gene networks for all samples, 25 genes had an average edge density of 479.2, which was relatively high compared to the whole network, which had an average edge density of 65.8.

The functional analysis of 25 prognostic genes was then performed using DAVID [26], and the terms of Benjamini *p*-value < 0.05, among the terms corresponding to the Biological Process among KEGG Pathway and GO, are presented in Appendix A. Of the total 38 terms, 31 had been studied in the literature for relevance to breast cancer, while 10 were relevant to the prognosis of breast cancer (Appendix A).

Furthermore, Kaplan–Meier analysis was performed on 25 prognostic genes. In total, 18 genes showed a *p*-value < 0.05 (Appendix A), and of these, nine genes (*APC*, *BRCA1*, *COPS5*, *FOXD1*, *MAPK10*, *MAPK14*, *NCOA3*, *PAX6*, *PLK1*, Appendix A) showed a low survival probability in the highly expressed group, among which two BRCA driver genes, APC and BRCA1, were included. In addition, nine genes (*BATF*, *BCL3*, *FLT3*, *JUND*, *MAX*, *NFATC1*, *RUNX1*, *TGIF1*, *XBP1*, Appendix A) showed high survival probability in the high-expressed group, among which two BRCA driver genes, *MAX* and *RUNX1*, were included. We validated 25 genes using KMplot web [27] and found that the *p*-values of 17 genes were <0.05 (Appendix A). Among 17 genes, 15 genes were also significant in our Kaplan-Meier analysis. We also validated 25 genes using Protein Atlas [28]. Among the 25 genes, *JUND* and *BCL3* were reported as prognostic for BRCA (*p*-value = 0.000061 and 0.00097, respectively), and were also significant in our Kaplan-Meier analysis (*p*-value = 0.0005 and 0.0025, respectively). In addition, 19 genes showed *p*-value < 0.05 in Protein Atlas. Among 19 genes, 15 genes also showed significant *p*-values in our Kaplan–Meier analysis (Appendix A).

In addition, we performed a correlation analysis with immune cells using TIMER [29] for 25 genes. We found many of them to have significant positive and/or negative correlations with ten T cells (Appendix A). Among them, *RUNX1* and *XBP1* showed generally negative correlations, while *IRF7*, *KAT2B*, *MAX*, and *NCOA3* showed generally positive correlations, and *BATF* and *FLT3* showed positive correlations in BRCA-Basal and BRCA-Her2. These results indirectly showed how the selected gene affected the prognosis.

Next, prognostic genes were selected for CESC, including BRCA, while 20 genes were identified in three out of the ten experiments and were selected as prognostic genes (Appendix A). Six of the twenty prognostic genes were selected more than four times. Unlike in BRCA, no known driver genes were included among the 20 prognostic genes. A subgraph of the 20 prognostic genes is shown in Figure 7. Similarly to BRCA, the average edge density of these 20 genes was higher than that of the entire network (357.2 vs. 65.8). *TBP* was selected seven times, whereas the proto-oncogene *MYC* was not selected at all. This may have been because *MYC* was a driver gene regardless of prognosis, whereas *TBP* may be a novel driver candidate related to *MYC* and the prognosis of CESC.

Similarly to BRCA, a functional analysis was performed here, with the results shown in Appendix A. Of the total 37 terms, 25 had previously been studied for their relevance to cervical squamous cell carcinoma, while 11 were relevant to the prognosis of cervical squamous cell carcinoma. Four KEGG pathways (human T-cell leukemia virus 1 infection, colorectal cancer, breast cancer, and pathways in cancer) and five GO terms, including the regulation of transcription from the RNA polymerase II promoter and DNA-templated regulation of transcription, were commonly enriched in BRCA and CESC. Among these common terms, the most unexpected was human T-cell leukemia virus 1 infection, which may share pathways related to the prognosis of CESC and BRCA.

Kaplan–Meier analysis was also performed on the 20 CESC prognostic genes. Twelve genes showed a *p*-value < 0.05 (Appendix A), and of these, eight genes (*ATF2*, *BRCA2*, *FAM120B*, *GTF2I*, *NUP58*, *PLAGL1*, *SKIL*, and *TBP*; Appendix A) showed a low survival probability in the high-expression group. Furthermore, four genes (*HLF*, *POU2F1*, *SOX10*, and *TCF7*; Appendix A) were associated with a high probability of survival in the high-expression group. Finally, we analyzed 20 genes using Protein Atlas [28]. Among the 20 genes, NUP58 was reported as prognostic for CESC (*p*-value = 0.00026, high expression is unfavorable), and was also significant in our Kaplan–Meier analysis (*p*-value = 0.0075). In addition, 10 genes (*SKIL*, *KDM5B*, *TCF7*, *RPS6KB2*, *PLAGL1*, *NFYA*, *HLF*, *TGFB1*, *PTTG1*, and *GTF2I*) showed *p*-value < 0.05 in Protein Atlas (Appendix A). Among 10 genes, five genes (*SKIL*, *TCF7*, *PLAGL1*, *HLF*, and *GTF2I*) also showed significant *p*-values in our Kaplan-Meier analysis. Similarly to BRCA, these results showed how the selected genes affected prognosis.

## 3. Discussion

In this study, it was demonstrated that patient-specific cancer driver genes could be used to predict cancer prognoses more accurately. Firstly, patient-specific gene networks were generated using the differences in gene expression between cancer and normal samples for cancer prognosis prediction. Subsequently, modified PageRank was used to generate the feature vectors. These feature vectors represented the impact or influence of genes on all genes in the patient-specific gene network. The feature vectors of the good- and poor-prognosis groups were subsequently used to train the deep feedforward network.

The proposed method generally outperformed the existing state-of-the-art methods in predicting the prognoses of 11 cancer types. In particular, the proposed method significantly outperformed DeepProg, which was the second-best method for BRCA, CESC, and PAAD, while outperforming DeepProg for five more cancer types. These results were relatively surprising, considering that the classification model of the proposed method was a simple deep feed-forward network, whereas DeepProg adopted a sophisticated deep-learning-based classification model. It was concluded that a good performance originated from the proper feature set, which was the driver gene score.

The novelty of the proposed method can be summarized as: (1) the novel patient-specific gene network generation scheme; and (2) the generation of a feature vector of driver gene scores, which is appropriate for cancer prognosis.

The advantage of the proposed method over previous methods is its high accuracy in the prediction of cancer prognosis and its ability to select proper prognostic genes. However, the running time of the proposed method is relatively slower than other methods because patient-specific gene networks are generated and processed for each sample.

The results are limited in that further validation through wet lab experiments is still necessary to confirm that the derived driver genes were actually driver genes. Alternatively, it can be indirectly inferred that driver genes were properly derived using known driver genes from the CGC or Intogen databases. However, the number of known driver genes was not sufficient for a meaningful statistical evaluation. For example, there were six known driver genes in 30 genes selected from BRCA, while there were no known driver genes in 25 genes from CESC. However, some of these 25 genes were clearly related to CESC and were likely to have been driver genes of CESC. The proposed method could be improved upon by developing a better patient-specific driver gene method, which will be investigated in our ongoing research paper.

## 4. Methods and Materials

### 4.1. Overview

A patient-specific gene network was first constructed, before generating feature vectors by scoring patient-specific genes using PageRank and then correcting patient-specific gene scores using somatic mutation data. For each cancer sample, a set of feature vectors was created by comparing it to the normal sample group, with a feature vector of the gene representing its influence on the genes in the patient-specific gene network. Therefore, these feature vectors could be used to identify cancer driver genes, while the feature vectors of the good and poor prognosis groups showed differences in the similarity of the potential driver genes for each group. The feature vectors of the good- and poor-prognosis groups were then used to train the deep feedforward network. Figure 8 presents a general overview of the proposed method.

### 4.2. Building the Patient-Specific Gene Network

First, the patient-specific gene network was constructed using gene expression data and integrated gene networks, which consisted of FI networks from Reactome [16] and gene regulation networks from RegNetwork [17] and TRRUST [18]. The proposed model was designed to search for prognosis-specific genes that have a significant influence on other genes. Therefore, the directions of all edges with direction were reversed, as in DawnRank.

A patient-specific gene network is represented by W, which is a weighted adjacency matrix. W is calculated using two matrices, A and Φ, as shown in Equation (1).
(1)W=A ⊗ Φ
where A is an n×n adjacent matrix of the integrated gene network calculated by Equation (2), while Φ is a matrix calculated based on R. Φ and R are calculated using Equations (3) and (4), respectively.
(2)$$A_{ij}=\{\begin{array}{c} 1,\;\;if\;gene_{i}\;\mathrm{and}\;gene_{j}\;\mathrm{are}\;\mathrm{linked}\;\mathrm{in}\;a\;\mathrm{gene}\;\mathrm{network}\\ 2,\;\;\mathrm{if}\;gene_{i}\;\mathrm{and}\;gene_{j}\;\mathrm{are}\;\mathrm{linked}\;\mathrm{and}\;gene_{i}\;\mathrm{or}\;gene_{j}\;\mathrm{has}\;\mathrm{somatic}\;\mathrm{mutation}\;\\ 0,\;\;\mathrm{if}\;gene_{i}\;\mathrm{and}\;gene_{j}\;\mathrm{are}\;\mathrm{not}\;\mathrm{linked}\;\mathrm{in}\;a\;\mathrm{gene}\;\mathrm{network} \end{array}$$
(3)$$R_{g}=\left|T_{rank}-N_{rank}\right|\;of\;gene\;g$$
(4)$$\Phi_{ij}=\left(R_{g_{i}}+R_{g_{j}}\right)\times min\left(R_{g_{i}},R_{g_{j}}\right)$$

In Equation (3), $T_{rank}$ is the rank of the expression value of a gene in a single cancer sample, while $N_{rank}$ is the rank of the average expression of a gene in all normal samples. The larger the value of the expression, the higher the rank. Additionally, $R_{g}$ represents the difference in the expression of gene g between the cancer and normal samples. A larger $R_{g}$ value indicates that gene g is more significant for the prediction of prognosis. In (4), $\Phi_{ij}$ is a weight calculated as the rank difference between $gene_{i}$ and $gene_{j}$. The term $min\left(R_{g_{i}},R_{g_{j}}\right)$ is multiplied since an edge with two significant nodes (genes) is better than an edge with only one significant gene. For example, given $R_{a}=6$, $R_{b}=1$, $R_{c}=3$, $R_{d}=4$, $\Phi_{ab}=7$, and $\Phi_{cd}=21$, although $(R_{a}+R_{b})=\left(R_{c}+R_{d}\right)$. An example of calculating R, Φ, and W is shown in Figure 9.

### 4.3. Calculation of Genetic Impact Scores

The score $S_{i}$ of $gene_{i}$ for each patient can be calculated using Equation (5).
(5)$$S_{i}=\left(1-d\right)f_{i}+d\times WS_{i-1},$$
where $f_{i}$ is the absolute value of the difference in $gene_{i}$ between a cancer sample and a group of normal samples. The damping factor d is expressed in Equation (6) as in DawnRank, where the number of incoming edges for each gene is $A_{sum}$.
(6)$$d=\frac{A_{sum}}{A_{sum}+3}$$

After a vector S is created for each cancer sample, a penalty is given for genes without genetic mutations, as shown in Equation (7), to correct the genetic score calculated in Section 2.3. for somatic cell mutations.
(7)$$S_{i}=\{\begin{array}{l} S_{i}\;\;\;\;\;\;\;\;\;\;\;\;,if\;gene_{i}\;has\;genetic\;variation\\ S_{i\;}\times p\;\;\;,otherwise\;\;\;\;\;\;\;\;\;\;\;\;\;\;\;\;\;\;\;\;\;\;\;\;\;\;\;\;\;\;\;\;\;\;\;\;\;\;\;\; \end{array},$$

In Equation (7), $S_{i}$ represents the genetic score of $gene_{i}$, while *p* is a penalty, which was set to 0.85 here, as in DawnRank.

The winning rate vector *V* is calculated and then used for training. If genes have a somatic mutation, its winning rate is calculated by comparing it with all other genes; otherwise, it is calculated by comparing it with genes that had only somatic mutations. An example is shown in Figure 10.

Considering that *f* is a vector of initial gene expression differences, and is propagated through the inversely directed gene network, *S* and *V* can be seen as vectors of the impacts or influences that genes have on the patient-specific gene network and can also be seen as the likelihood that genes can be cancer driver genes for a specific cancer sample.

Subsequently, only genes with significant *p*-values were selected by performing a *t*-test on the gene expression data of the good- and poor-quality groups. The prognostic prediction model was trained using a deep neural network with only genes having significant *p*-values in *V*.

## Figures and Tables

**Figure 1 ijms-24-06445-f001:**
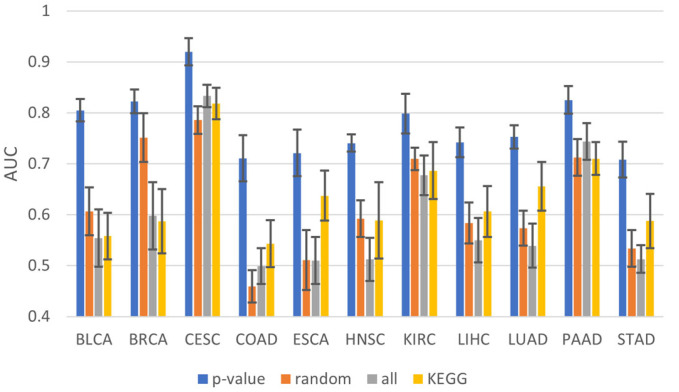
Average AUC of 10 independent tests for varying gene selection methods.

**Figure 2 ijms-24-06445-f002:**
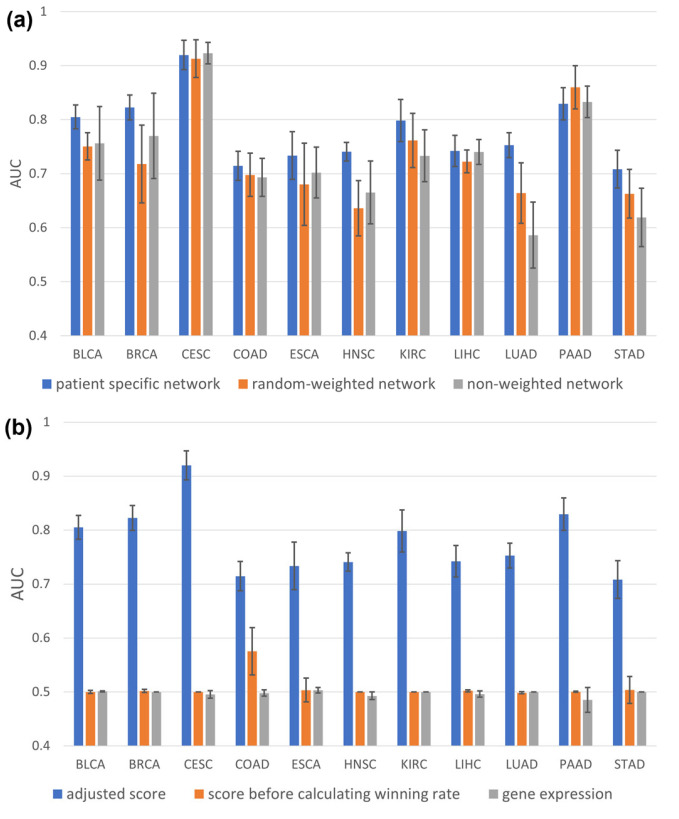
Model configuration comparison: (**a**) Average AUC of 10 independent tests of each cancer for network configuration comparison. (**b**) Average AUC of 10 independent tests of each cancer for score generation comparison.

**Figure 3 ijms-24-06445-f003:**
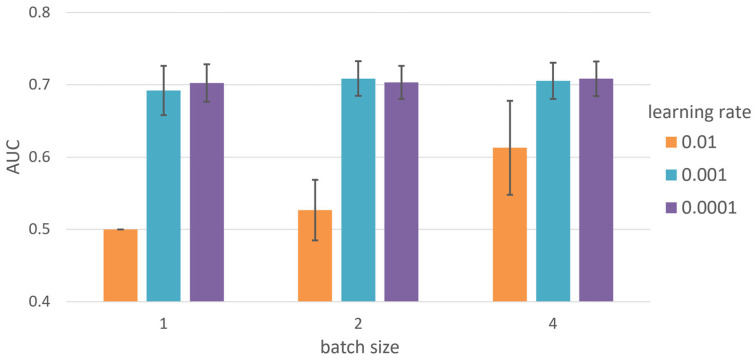
Average AUC of 10 independent tests for varying batch size and learning rate of DNN.

**Figure 4 ijms-24-06445-f004:**
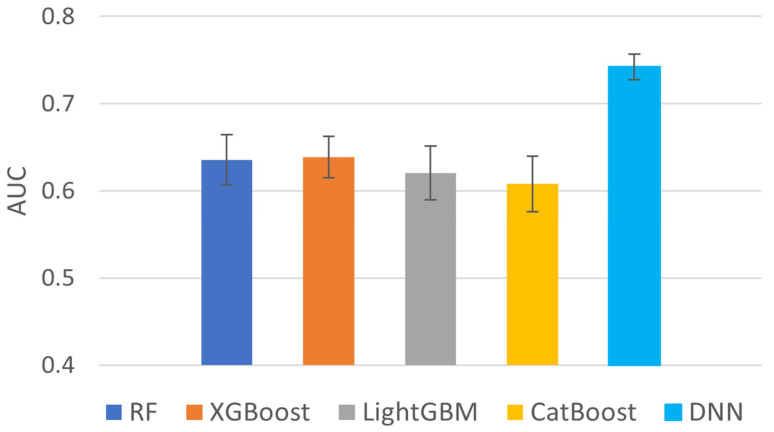
Average AUC of 10 independent tests for various classification models.

**Figure 5 ijms-24-06445-f005:**
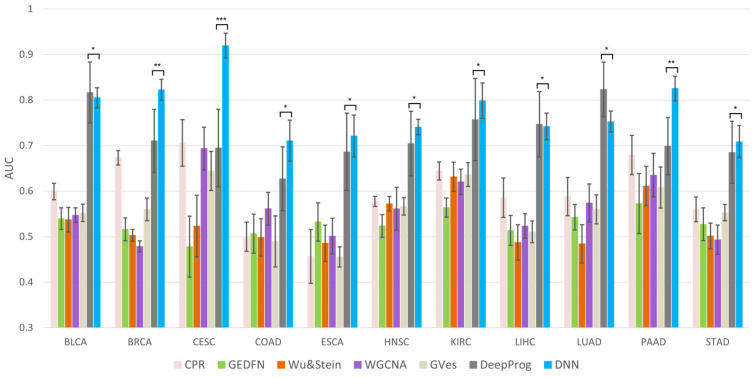
Comparison of average AUC from 10 independent tests for each cancer. The * mark above the graph shows the multiple testing results of DNN and DeepProg. (* *p*-value > 0.05, ** *p*-value < 0.05, *** *p*-value < 0.001).

**Figure 6 ijms-24-06445-f006:**
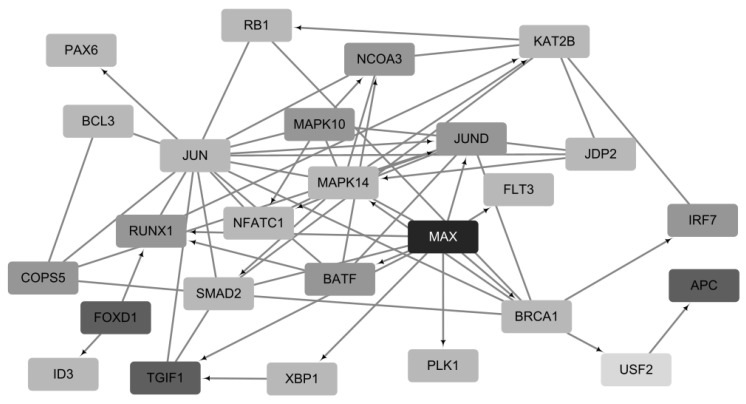
Network for genes derived more than thrice out of the top 25 genes in BRCA. MAX was selected six times, three genes (*APC*, *FOXD1* and *TGIF1*) were selected five times, seven genes (*COPS5*, *RUNX1*, *BATF*, *MAPK10*, *NCOA3*, *JUND* and *IRF7*) were selected more than four times, and the remaining 13 genes were selected three times out of ten times. *USF2* (selected twice) was not included in top 25 genes.

**Figure 7 ijms-24-06445-f007:**
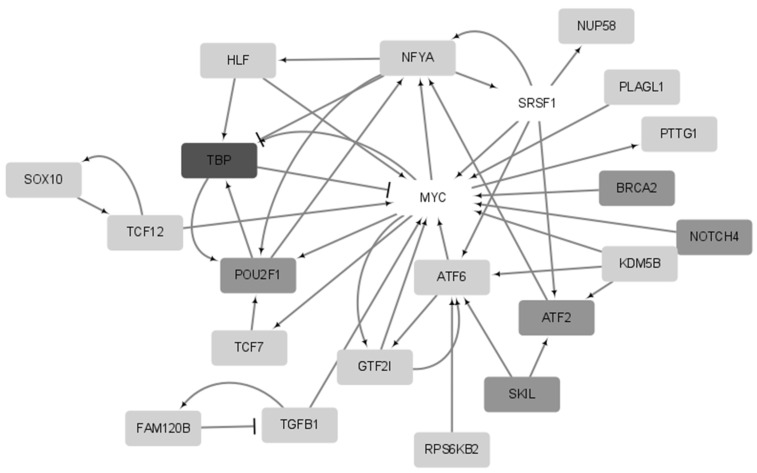
Network for genes derived more than three times out of the top 20 genes in CESC. *TBP* was selected seven times, five genes (*BRCA2*, *ATF2*, *POU2F1*, *NOTCH4* and *SKIL*) were selected four times, and the remaining fourteen genes were selected three times out of 10. *MYC* and *SRSF1* (not selected) were not included in the 20 genes, but are shown in the figure.

**Figure 8 ijms-24-06445-f008:**
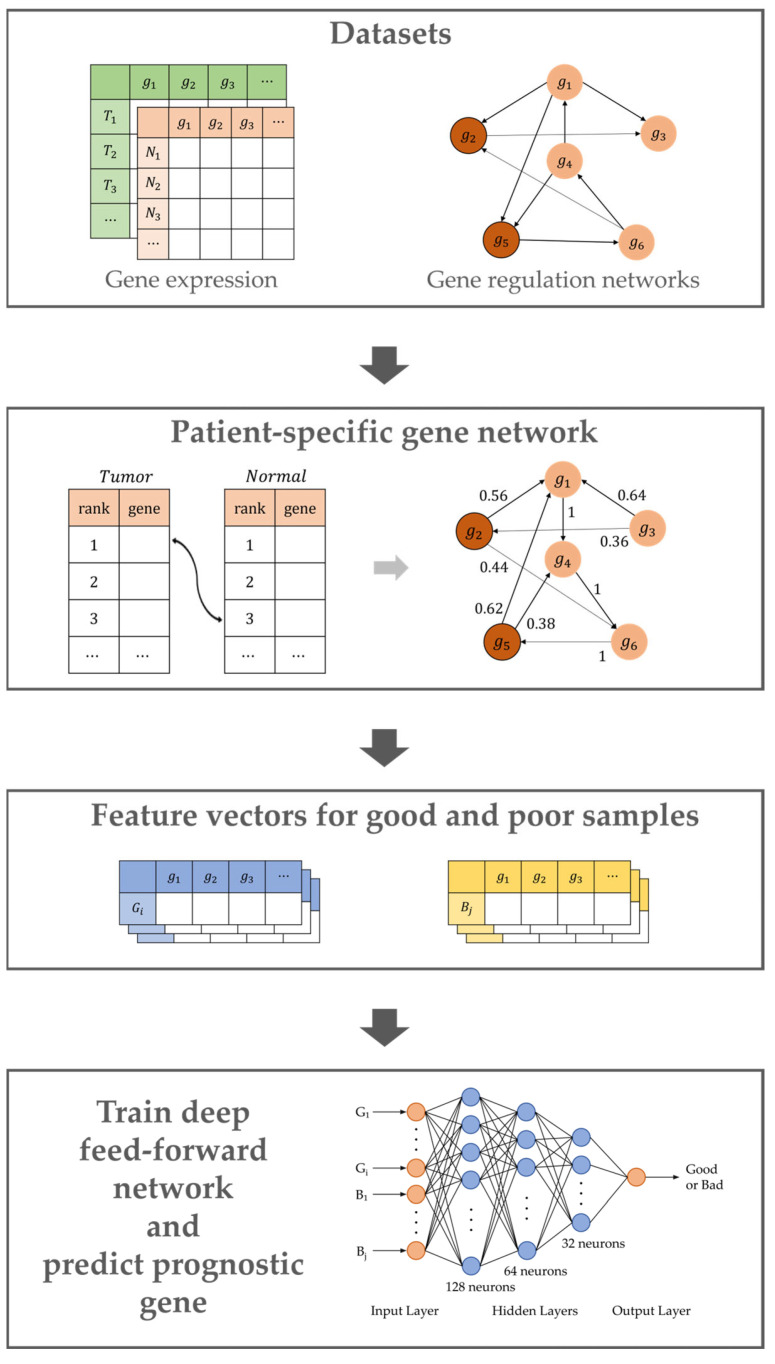
Overview of the proposed method.

**Figure 9 ijms-24-06445-f009:**
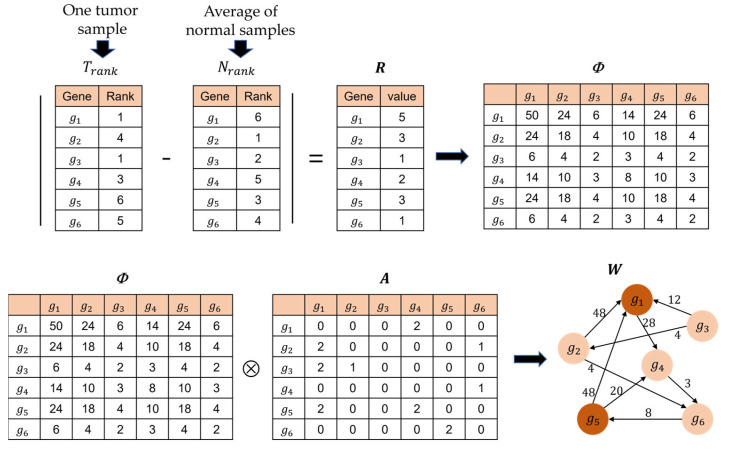
Example for building a patient-specific genetic network. *g*_1_ and *g*_5_ have somatic mutations.

**Figure 10 ijms-24-06445-f010:**
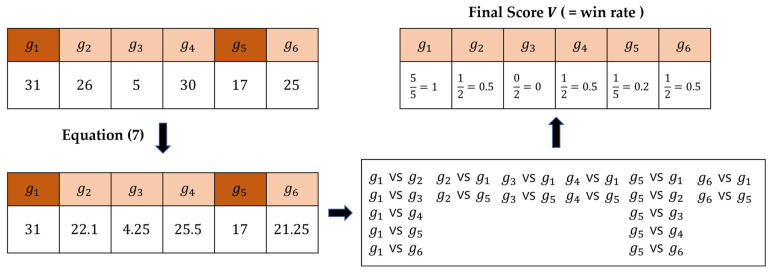
Patient-specific gene score correction process. *g*_1_ and *g*_5_ have somatic mutations.

**Table 1 ijms-24-06445-t001:** Data description.

Cancer	Criteria for Label	Number of Good Samples	Number of Bad Samples	Number of Normal Samples	Number of Genes
BLCA	2 years	73	83	19	15,779
BRCA	5 years	90	63	113	15,557
CESC	4 years	25	23	3	15,506
COAD	3 years	35	32	41	15,299
ESCA	1 years	33	30	11	15,984
HNSC	2 years	99	120	44	15,933
KIRC	4 years	66	51	72	15,782
LIHC	2 years	78	55	50	14,965
LUAD	2 years	67	64	59	15,417
PAAD	1 years	34	27	4	16,024
STAD	1 years	63	45	35	15,638

## Data Availability

https://github.com/SYL320/accurate-prediction-prognosis.git (accessed on 21 February 2023), TCGA, https://portal.gdc.cancer.gov/ (accessed on 21 February 2023), Reactome, https://reactome.org/download-data (accessed on 21 February 2023), RegNetwork, https://regnetworkweb.org/download.jsp (accessed on 21 February 2023), TRRUST, https://www.grnpedia.org/trrust/downloadnetwork.php (accessed on 21 February 2023).

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
