# Peer review of "Accurate Prediction of Cancer Prognosis by Exploiting Patient-Specific Cancer Driver Genes"

_ijms, 2023, doi:10.3390/ijms24076445_

Round 1

Reviewer 1 Report

The authors demonstrated that patient-specific cancer driver genes could be used to predict cancer prognoses more accurately. Patient-specific gene networks were generated using the differences in gene expression between cancer and normal samples for cancer prognosis prediction. Next, modified PageRank was used to generate the feature vectors. These feature vectors represented the impact or influence of genes on all genes in the patient-specific gene network. The feature vectors of the good and poor-prognosis groups were subsequently used to train the deep feedforward network. Albeit, I consider these findings to provide new insight into cancer-related fields, I still have some suggestions.

1, Most figures are highly professional, however, the authors should guide the readers to the meaning of the images appropriately; otherwise, it is likely to cause misunderstandings. Therefore, I suggest that the author consider revising these figure legends again.

2, In figure 8, the author showed the network for genes derived more than thrice out of the top 25 genes in BRCA. It is worth exploring and validating their data via kmplot (https://kmplot.com) (PMID: 34309564, 34202528, 34834441)

3, So far, the tumor infiltrates immune cells and is vital for patient survival. Therefore, it is worth validating their data correlated with immune cells by using the "TIMER" (http://timer.cistrome.org) analysis tool (PMID: 32442275, 36769082, 34329194).

4, In Figure 9, the authors showed the network for genes derived more than three times out of the top 20 genes in CESC. Since Connectivity Map (CMap) can be used to discover the mechanism of action of small molecules, functionally annotate genetic variants of disease genes, and inform clinical trials. It would be fascinating if these data could be correlated with other clinical databases. Therefore, I suggest the authors can validate their data via CMap or proteinatlas, and discuss these methodologies and literature as well as the validated data for cancer recurrence or metastasis in the manuscript (PMID: 29195078, 34512160, 32064155)

5, There are few typo issues for the authors to pay attention to; please also unify the writing of scientific terms. “Italic, capital”? The font is too small for some of the current figures; meanwhile, the manuscript also needs English proofreading.

Author Response

We (the authors) would like to thank you for your time and effort to review our paper. Your comments are invaluable in improving the quality of our paper.

  1. Most figures are highly professional, however, the authors should guide the readers to the meaning of the images appropriately; otherwise, it is likely to cause misunderstandings. Therefore, I suggest that the author consider revising these figure legends again.

Answer: Thank you for your comment. We modified legends of Figures 4,5 and 6.

  1. In figure 8, the author showed the network for genes derived more than thrice out of the top 25 genes in BRCA. It is worth exploring and validating their data via kmplot (https://kmplot.com) (PMID: 34309564, 34202528, 34834441)

Answer: Thank you for your valuable and constructive suggestion. We validated 25 genes of Breast cancer using KMplot and found that p-values of 17 genes were < 0.05. Among 17 genes, 15 genes were also significant in our Kaplan-Meyer analysis. We modified the manuscript accordingly and included the results in Supplementary figure 3.

  1. So far, the tumor infiltrates immune cells and is vital for patient survival. Therefore, it is worth validating their data correlated with immune cells by using the "TIMER" (http://timer.cistrome.org) analysis tool (PMID: 32442275, 36769082, 34329194).

Answer: Thank you for your valuable and constructive suggestion. We also validated 25 genes of Breast cancer using TIMER. We found many of them to have significant positive and/or negative correlation with ten T-cells. Among them, RUNX1 and XBP1 showed generally negative correlations, while IRF7, KAT2B, MAX and NCOA3 showed generally positive corelations, and BATF and FLT3 showed positive correlations in BRCA-Basal and BRCA-Her2. We modified the manuscript accordingly and included the results in Supplementary figure 4.

  1. In Figure 9, the authors showed the network for genes derived more than three times out of the top 20 genes in CESC. Since Connectivity Map (CMap) can be used to discover the mechanism of action of small molecules, functionally annotate genetic variants of disease genes, and inform clinical trials. It would be fascinating if these data could be correlated with other clinical databases. Therefore, I suggest the authors can validate their data via CMap or proteinatlas, and discuss these methodologies and literature as well as the validated data for cancer recurrence or metastasis in the manuscript (PMID: 29195078, 34512160, 32064155)

Answer: Thank you for your valuable and constructive suggestion. We tried validation of top 20 genes in CESC using Protein Atlas. Among the 20 genes, NUP58 was reported as prognostic for CESC (p-value = 0.00026, high expression is unfavorable), and was also significant in our Kaplan-Meier analysis (p-value = 0.00026). In addition, 10 genes (SKIL, KDM5B, TCF7, RPS6KB2, PLAGL1, NFYA, HLF, TGFB1, PTTG1 and GTF2I) showed p-value < 0.05 in Protein Atlas. Among 10 genes, five genes (SKIL, TCF7, PLAGL1, HLF and GTF2I) also showed significant p-values in our Kaplan-Meier analysis. These results were included in the manuscript and supplementary table 3.

We also validated top 25 genes in BRCA using Protein Atlas. Among the 25 genes, JUND and BCL3 were reported as prognostic for BRCA (p-value = 0.000061 and 0.00097, respectively), and were also significant in our Kaplan-Meier analysis (p-value = 0.0005 and 0.0025, respectively). In addition, 19 genes showed p-value < 0.05 in Protein Atlas. Among 19 genes, 15 genes also showed significant p-values in our Kaplan-Meier analysis. These results were included in the manuscript and supplementary table 3.

We also tried CMap, but were not able to get meaningful results, so did not include them in the manuscript.

  1. There are few typo issues for the authors to pay attention to; please also unify the writing of scientific terms. “Italic, capital”? The font is too small for some of the current figures; meanwhile, the manuscript also needs English proofreading.

Answer: Thank you for your comment. We carefully modified and proofread the manuscript.

Reviewer 2 Report

This manuscript entiled "Accurate Prediction of Cancer Prognosis by Exploiting Patient Specific Cancer Driver Genes" by Lee S. et al. indicated prediction models of various cancer types with the disease-associated gene networks uisng machine learning. This manuscript is very important and interesting in this field. But, some corrections may be needed. In figure 4 to 7, AUC is used as evaluation index. It was better to add other evaluation indexes, such as balanced accuracy, f-value, PR_AUC, and matthews correlation coefficient. As for the neural network used in this study, it was better to show the structure as figure. In figure 5, as for the performances, it was better to calculate the statistical significances by multiple testing. In 3.4 comparison on different machine learning methods, it was better to add results by other machine learnings, such as random forest, vgboost, lightgbm, and catboost etc. In discussion section, it was better to add advantages/disadvantages of this method with previous reports, and referrences. In addition, it was better to conclusion section to show the novelity in this study.  

Author Response

We (the authors) would like to thank you for your time and effort to review our paper. Your comments are invaluable in improving the quality of our paper.

  1. In figure 4 to 7, AUC is used as evaluation index. It was better to add other evaluation indexes, such as balanced accuracy, f-value, PR_AUC, and matthews correlation coefficient.

Answer: Thank you for your valuable comment. We used the suggested evaluation indexes and modified the manuscript accordingly. The results were included in Supplementary figure 1.

  1. As for the neural network used in this study, it was better to show the structure as figure.

Answer: Thank you for your constructive comment. We modified Figure 1 to show the neural network structure.

  1. In figure 5, as for the performances, it was better to calculate the statistical significances by multiple testing.

Answer: Thank you for your valuable comment. We calculated the statistical significances and showed them in Figure 8.

  1. In 3.4 comparison on different machine learning methods, it was better to add results by other machine learnings, such as random forest, vgboost, lightgbm, and catboost etc.

Answer: Thank you for your constructive comment. We compared the neural network with suggested machine learning methods and added the results as Figure 7 in section 3.3.

  1. In discussion section, it was better to add advantages/disadvantages of this method with previous reports, and referrences.

Answer: Thank you for your valuable comment. We added advantages and disadvantages of the proposed method in the Discussion.

  1. In addition, it was better to conclusion section to show the novelty in this study.

Answer: Thank you for your constructive comment. We added novelty of the proposed method in the Discussion.

Round 2

Reviewer 2 Report

This manuscript entitled "Accurate Prediction of Cancer Prognosis by Exploiting Patient Specific Cancer Driver Genes" by Lee S. et al., has benn corrected clearly according to reviewers comments. It will be very important and significant report in this field.